# Elucidation of Focal Adhesion Kinase as a Modulator of Migration and Invasion and as a Potential Therapeutic Target in Chronic Lymphocytic Leukemia

**DOI:** 10.3390/cancers14071600

**Published:** 2022-03-22

**Authors:** Thomas A. Burley, Andrew Hesketh, Giselda Bucca, Emma Kennedy, Eleni E. Ladikou, Benjamin P. Towler, Simon Mitchell, Colin P. Smith, Christopher Fegan, Rosalynd Johnston, Andrea Pepper, Chris Pepper

**Affiliations:** 1Department of Clinical and Experimental Medicine, Brighton and Sussex Medical School, Falmer, Brighton BN1 9PX, UK; t.burley@bsms.ac.uk (T.A.B.); e.m.kennedy@bsms.ac.uk (E.K.); e.e.ladikou@bsms.ac.uk (E.E.L.); b.towler2@bsms.ac.uk (B.P.T.); s.a.mitchell@bsms.ac.uk (S.M.); c.pepper@bsms.ac.uk (C.P.); 2School of Applied Sciences, University of Brighton, Brighton BN2 4GJ, UK; a.hesketh@brighton.ac.uk (A.H.); g.bucca@brighton.ac.uk (G.B.); c.p.smith@surrey.ac.uk (C.P.S.); 3Department of Haematology, Brighton and Sussex University Hospital Trust, Brighton BN2 5BE, UK; rosalynd.johnston1@nhs.net; 4Department of Nutritional Sciences, Faculty of Health and Medical Sciences, School of Biosciences and Medicine, University of Surrey, Guildford GU2 7YH, UK; 5Division of Cancer and Genetics, School of Medicine, Cardiff University, Heath Park, Cardiff CF14 4XN, UK; fegancd1@cardiff.ac.uk

**Keywords:** chronic lymphocytic leukemia, FAK, TLR9, migration, transcriptomics, miRNomics, microenvironment

## Abstract

**Simple Summary:**

Despite the successful introduction of targeted therapies, Chronic Lymphocytic Leukemia (CLL) remains incurable. This is thought to be partially due to the pro-survival and anti-apoptotic signaling that CLL cells receive from the lymph node microenvironment. Therefore, inhibition of CLL migration into the lymph nodes is an attractive therapeutic option. Here, our aim was to gain a further understanding of what transcriptomic and miRNomic changes drive CLL migration and, from this, select promising therapeutic targets. We identified focal adhesion kinase (FAK) as one such potential target and demonstrated that inhibition of FAK in primary CLL samples effectively reduces both CXCL12 induced migration and invasion in vitro. Successful inhibition of CLL migration could increase the sensitivity of CLL cells to the currently used targeted therapeutics and therefore improve patient outcomes.

**Abstract:**

The retention and re-migration of Chronic Lymphocytic Leukemia cells into cytoprotective and proliferative lymphoid niches is thought to contribute to the development of resistance, leading to subsequent disease relapse. The aim of this study was to elucidate the molecular processes that govern CLL cell migration to elicit a more complete inhibition of tumor cell migration. We compared the phenotypic and transcriptional changes induced in CLL cells using two distinct models designed to recapitulate the peripheral circulation, CLL cell migration across an endothelial barrier, and the lymph node interaction between CLL cells and activated T cells. Initially, CLL cells were co-cultured with CD40L-expressing fibroblasts and exhibited an activated B-cell phenotype, and their transcriptional signatures demonstrated the upregulation of pro-survival and anti-apoptotic genes and overrepresentation of the NF-κB signaling pathway. Using our dynamic circulating model, we were able to study the transcriptomics and miRNomics associated with CLL migration. More than 3000 genes were altered when CLL cells underwent transendothelial migration, with an overrepresentation of adhesion and cell migration gene sets. From this analysis, an upregulation of the FAK signaling pathway was observed. Importantly, *PTK2* (FAK) gene expression was significantly upregulated in migrating CLL cells (*PTK2* Fold-change = 4.9). Here we demonstrate that TLR9 agonism increased levels of p-FAK (*p* ≤ 0.05), which could be prevented by pharmacological inhibition of FAK with defactinib (*p* ≤ 0.01). Furthermore, a reduction in CLL cell migration and invasion was observed when FAK was inhibited (*p* ≤ 0.0001), supporting a role for FAK in both CLL migration and tissue invasion. When taken together, our data highlights the potential for combining FAK inhibition with current targeted therapies as a more effective treatment regime for CLL.

## 1. Introduction

Chronic Lymphocytic Leukemia (CLL) is the most common leukemia in Western countries, characterized by the clonal expansion of B cells in the lymphoid organs, such as the bone marrow and lymph nodes [1]. Despite improvements in patient survival, due to the development of novel targeted therapies [2,3], CLL remains incurable. This may partially be due to the sequestering of CLL cells in protective niches, such as the lymph nodes.

It was long thought that CLL development was primarily caused by the slow accumulation of tumor cells due to failed apoptosis [4,5]. However, studies have since shown that there is significant CLL cell proliferation and turnover every day [6]. CLL cells in the peripheral blood are largely arrested in the G0/G1 phase of the cell cycle, and their survival is highly dependent on interactions with the microenvironment, as emphasized by the rapid apoptosis observed during in vitro CLL cell culture [7]. In the peripheral blood, CLL cell interaction with endothelial cells has been shown to stimulate survival but not proliferation [8,9], with CLL proliferation appearing to be restricted to proliferation centers in the lymphoid tissues [10]. These lymphoid tissues also provide a protective niche to help CLL cells avoid destruction by therapeutic agents [11]. Therefore, migration and recirculation of CLL cells between the lymphoid niches and peripheral blood, where the malignant cells receive pro-survival and pro-proliferation signaling via the B cell receptor (BCR) as well as stromal and T cell interaction via proteins such as CD40L, are critical factors in determining CLL progression and treatment resistance [12,13,14].

Targeting the BCR pathway with BTK inhibitors, such as ibrutinib, has revolutionized the treatment of CLL, producing a durable response in many patients [15,16,17]. However, despite a high overall response rate, only a small proportion of patients have a complete response, which highlights the need for combination strategies to improve survival outcomes in CLL [18,19,20]. Most patients that are treated with ibrutinib experience lymphocytosis, due to lymphocyte egress from the bone marrow, spleen, and lymph nodes into the peripheral blood [21]. For this reason, combinatorial targeting of CLL migration alongside ibrutinib could improve treatment efficacy by maximizing egress and minimizing ingress, resulting in a more complete inhibition of proliferation and diminished stroma-mediated protection of CLL cells.

Cell migration is a complex process, with the signaling mechanisms that control CLL trafficking into the lymphoid niches remaining largely unknown. It has been described that stromal cells in the lymphoid niche secrete chemokines such as CXCL12 and CXCL13, which bind to their corresponding receptors on CLL cells (CXCR4 and CXCR5), resulting in leukemic cell chemotaxis towards the lymphoid niche [22]. However, it is clear that CLL transendothelial migration is dependent on adhesion molecules such as the integrin VLA4 (CD49d) [23] and selectin CD62L [24]. In fact, CD49d is a powerful prognostic marker in CLL [25,26], with high expression associated with aggressive disease, emphasizing the importance of CLL lymph node migration on tumor survival. With this being the case, inhibition of CLL migration is an attractive therapeutic approach.

We have previously shown phenotypic differences between non-migrating and migrating CLL cells [27], with the latter having a striking similarity to lymph node resident CLL cells [28]. The primary objective of this study was to investigate the differential transcriptomics and miRNomics between migrating, non-migrating, and lymph node resident CLL cells, in order to further elucidate the process of CLL trafficking and thereby potentially identify novel therapeutic targets. We performed this by utilizing an in vitro circulating model system that more accurately recapitulates the capillary beds in comparison to traditional 2D (static) culture methods [27]. In addition, we mimicked lymph node associated interactions using the established model of CD40L transfected fibroblasts [9,29,30]. This approach allowed us to make a three-way comparison of the transcriptomes of CLL cells. From the transcriptomic analysis, we observed an enrichment for cell activation and migration pathways in the actively trafficking CLL cells in our model when compared to those that remained in the circulation. We then identified a promising target protein, focal adhesion kinase (FAK), and assessed the impact of its inhibition on CLL migration.

## 2. Materials and Methods

### 2.1. Sample Collection and Processing

Twenty-six patients from the Royal Sussex County Hospital and four patients from the University Hospital of Wales with treatment-naïve CLL were included in the study. Peripheral blood samples were obtained from CLL patients with informed consent in accordance with the Declaration of Helsinki, and ethical approvals were granted by the South East Wales Research Ethics Committee (02/4806) and the Central Bristol Research Ethics Committee (17/SW/0263). Appendix A shows the immunophenotypic characteristics of the patient samples used in each experiment.

### 2.2. In Vitro Circulatory System

A hollow fiber bioreactor system (Appendix A) (FiberCell Systems Inc., New Market, MD, USA) was adapted from the method described by Walsby et al. [27]. The insides of the Polysulfone hollow fibers were coated with gelatin (0.2%) to allow the adhesion of human umbilical vein endothelial cells (HUVECs) (1 × 10^7^ cells). CXCL12 (100 ng/mL) (BioLegend, San Diego, CA, USA) was added to the ‘Extra-vascular space’—the space outside of the hollow fibers—to encourage migration. CLL cells were introduced into the hollow fibers via one of the access ports and circulated around the system for 24 h. Circulating CLL cells were obtained by extracting 5 mL of media from the system. Migratory CLL cells were collected from the hollow fibers by flushing with PBS and then trypsinizing (0.25%, Sigma-Aldrich, St. Louis, MO, USA) the cartridge. CLL cells were then positively selected using the EasySep™ Human CD19 Positive Selection Kit (Stemcell Technologies, Vancouver, BC, Canada). Further details can be found in the Appendix A.

### 2.3. CD40L Fibroblast Co-Culture

NIH/3T3 murine fibroblasts transfected with human CD40L were seeded at 10^5^/mL in 24-well plates in RPMI-1640 and incubated overnight to allow cells to adhere. The next day, PBMCs from CLL patients (2 × 10^6^/mL) were cultured alone or added to the CD40L-expressing fibroblasts as previously described [9]. CLL cells were harvested after 4 h of culture. Subsequently, the expression of markers associated with cellular migration and/or activation (CD38, CD69, CD62L, and CXCR4), were quantified on CD19+/CD5+ CLL cells using mean fluorescent intensity (MFI) values. Labeling with fluorescent antibodies was carried out according to the antibody manufacturer’s instructions (Biolegend; the antibody panels used are detailed in Appendix A). For each sample, 10,000 events were acquired and compensated on a Cytoflex LX flow cytometer using CytExpert software (Beckman Coulter, Brea, CA, USA).

### 2.4. RNA-Sequencing and Analysis

RNA was extracted using the RNeasy Micro Kit (Qiagen, Hilden, Germany) as per the manufacturer’s instructions. Strand-specific RNA-seq libraries were prepared from the CLL cell paired samples using the NEBNext^®^ Ultra™ II Directional RNA Library Prep Kit for Illumina^®^ New England Biolabs according to manufacturer’s instructions and paired-end sequenced (2 × 75 cycles) using the Illumina NextSeq500 (Illumina, San Diego, CA, USA). Approximately 20 million sequencing read pairs were obtained per sample. Transcripts in the human genome (hg38) were quantified from the paired-end reads using kallisto [31], and gene-level count data were processed and analyzed for differential expression using DESeq2 [32]. XenofilteR was utilized to remove any mouse sequence reads from the co-culture experiment [33]. Over-representation analysis (ORA) was performed using WebGestalt. Signalling pathway networks displaying the differential expression data were created using Cytoscape v3.7.2.

### 2.5. MiRNA-Seq and Analysis

MiRNA-seq libraries were generated with the SMARTer smRNA-Seq Kit for Illumina (Takara Bio Europe, Gothenburg, Sweden). Small RNA-seq was performed using NextSeq500 Illumina (1 × 75 cycles), yielding 10–33 million single-end reads per sample. Reads were adapter trimmed using cutadapt v2.1 [34] with a command based on the kit manufacturers’ recommendations (cutadapt -m 15 -u 3 -max-n 0.9 -a AAAAAAAAAA), then processed to quantify human (hg38) miRNAs obtained from miRBase v22.1 using MiRDeep2 [35]. Briefly, this discards reads <18 nucleotides in length, collapses reads to uniqueness, and quantifies collapsed reads mapping to the hg38 genome. The miRNA count data were processed and analyzed for differential expression using DESeq2 [35].

### 2.6. Quantitative Real-Time PCR (qRT-PCR) of FAK (PTK2) RNA Expression

For the quantitative analysis of FAK RNA expression, the *PTK2* (Hs01056457_m1) and 18S rRNA Endogenous Control TaqMan (Hs99999901_s1, Applied Biosystems, Waltham, MA, USA) gene expression assays were used. Further details can be found in the Appendix A.

### 2.7. Apoptosis/p-FAK Immunostaining

CLL PBMCs were seeded at 5 × 10^5^ cells/200 μL of complete media (Media199, 10% fetal calf serum (FCS), penicillin/streptomycin and 5 ng/mL interleukin-4 [RayBiotech, Peachtree Corners, GA, USA]). Cells were cultured ±1 μM ODN2006 (TLR9 agonist; InvivoGen, San Diego, CA, USA) and incubated for 24 h. For FAK inhibition, CLL cells were pre-incubated for 2 h with Defactinib (Selleck Chemicals, Houston, TX, USA) at a range of concentrations (0.5 µM–5 µM). After 24 h, the amount of apoptosis was determined using a FITC-labeled Annexin V Apoptosis Detection Kit with 7-AAD (BioLegend). The p-FAK levels in the CLL cells were measured by intracellular staining with a PE-conjugated anti-p-FAK antibody (BD Biosciences, Franklin Lakes, NJ, USA), after fixing and permeabilizing the cells using the True-Phos™ kit (BioLegend) according to the manufacturer’s instructions. The MFI values for p-FAK were determined in gated CD19+/CD5+ CLL cells using a Cytoflex LX flow cytometer (Beckman Coulter).

### 2.8. Migration Assay

CLL PBMCs were seeded at 5 × 10^5^ cells/200 μL of complete media (Media199, 10% fetal calf serum (FCS), penicillin/streptomycin and 5 ng/mL interleukin-4 [RayBiotech]). Cells were cultured ±1 μM ODN2006 (TLR9 agonist; InvivoGen) ±1 μM defactinib (Selleck Chemicals) and incubated overnight. Transwell migration assays were performed using polycarbonate transwell inserts (5 μm pores) in 24-well plates (Corning Inc., Corning, NY, USA). A total of 600 μL complete media + 100 ng/mL CXCL12 (BioLegend) were added to the basolateral chambers, and PBMC (200 μL complete media) from CLL patients were transferred into the apical chambers and incubated for 4 h. Migrated CD19+/CD5+ CLL cells were quantified by volumetric counting using a Cytoflex LX flow cytometer (Beckman Coulter).

### 2.9. Invasion Assay

Matrigel-coated nucleopore filter inserts in a 24-well transwell chamber (Corning) were used for the invasion assays. Cells were seeded at a density of 5 × 10^5^ cells per well in Medium 199 supplemented with 10% FCS and IL-4 (5 ng/mL). PBMCs from six patients were pre-treated for 2 h with a range of defactinib concentrations (0.5 µM, 1 µM, 5 µM) or with an equivalent amount of DMSO. The cells were then added to the apical chamber of the invasion assay plate for 24 h (in defactinib + media) and encouraged to invade towards a CXCL12 gradient created by adding 700 µL complete media, supplemented with 10 ng/mL CXCL12 (R&D Systems, Minneapolis, MN, USA), into the basolateral chamber. Invading CD19+/CD5+ cells were quantitated by volumetric counting using the Cytoflex LX flow cytometer (Beckman Coulter).

### 2.10. Synergy between Defactinib and Ibrutinib

The potential synergy between defactinib in combination with ibrutinib was determined in primary CLL cells (*n* = 3). For this preliminary study, the cells were treated with three concentrations (0.5, 1, 2.5 µM) of each drug individually overnight (20 h) and in combination at a fixed molar ratio of 1:1. Cells were stained with Annexin V FITC/7-AAD and then analyzed on Cytoflex LX flow cytometer (Beckman Coulter). Transwell migration assays were performed (4 h incubation), and migrated cells were quantified by volumetric counting. The expected drug combination responses were calculated based on the Bliss reference model using SynergyFinder (https://synergyfinder.fimm.fi, accessed on 9 March 2022). Missing drug molar ratios were imputed before running the analysis. Bliss scores >10 strongly suggest synergistic interactions.

### 2.11. Statistics

Statistical analyses were performed by using GraphPad Prism 7.0 (GraphPad Software, San Diego, CA, USA) and SPSS. Unless otherwise stated, results are presented as mean ± standard deviation, and statistical significance was determined by using a paired, unpaired students *t*-test or one-way ANOVA. Differences were considered statistically significant when *p* ≤ 0.05.

## 3. Results

### 3.1. CLL 2D Cell Culture with CD40L Fibroblasts Upregulates Pro-Survival and Anti-Apoptotic Gene Sets

To mimic lymph node resident CLL cells, we used CD40L transfected fibroblasts [9,29,30]. Previously, our group reported that co-culture of CLL cells with CD40L-fibroblasts induced a consistent change in CLL phenotype, including increased expression of CD69 [9]. We, therefore, used CD69 as a marker of activation for this CLL co-culture model. Here, we showed that after just 4 h, all CLL cells had an activated phenotype as indicated by significantly increased CD69 expression (*p* ≤ 0.001; Appendix A). In contrast to the findings by Pasikowska et al., using lymph node fine needle aspirates, CD38 and CXCR4 were not significantly altered in our model system (Appendix A) [28].

We next investigated how CD40L-fibroblast stimulation altered CLL transcriptomics. This co-culture system, to some extent, replicated the lymph node interaction between CLL cells and activated T cells which drive tumor proliferation. CLL cells were either cultured alone or co-cultured with CD40L-expressing fibroblasts, and the transcriptional signatures from eight paired CLL patient samples were compared. After performing analysis using DESeq2, 1372 differentially expressed genes (*p*.adjust ≤ 0.05 and fold-change ≥ 1.5) were identified. Gene ontology and over-representation analysis (ORA) on the cell processes database was performed using WebGestalt (http://webgestalt.org, accessed on 21 November 2021). The analysis confirmed that co-cultured CLL cells had increased expression of genes associated with cytokine stimulation and, importantly, pro-survival and anti-apoptotic signatures, including *BCL-2* members, (0.59 log_2_FC), *BCL2L1* (3.5 log_2_FC), and *BCL2A1* (3.5 log_2_FC) (Figure 1A). ORA analysis, using the KEGG database, highlighted a prominent over-representation of the NF-κB signaling pathway and the TNF signaling pathway, consistent with CD40/CD40L interaction (Figure 1B). The differential expression data were input into a Cytoscape representation of known CD40L downstream pathways, which revealed a consistent upregulation of many of the relevant RNAs, e.g., (*CD40*, *TRAFs*) and subsequent activation of NF-κB (Appendix A), emphasizing the importance of the microenvironment on CLL survival and proliferation.

### 3.2. Comparative RNA-Sequencing of CLL 2D Cell Culture with CD40L-Expressing Fibro Blasts Produced Distinct Differential Expression Profiles to Those of Migratory CLL Cells from the In Vitro System

We have previously demonstrated that migrated CLL cells have a very similar phenotype to those that reside in the lymph node [28]. To compare the transcriptomic signatures between CLL cells from the CD40L-expressing co-culture system and the ones differentially expressed in the migrated versus circulatory cells from the dynamic circulating cell culture system, circulatory and migratory CLL cells from 10 patients were isolated from the system and differential gene expression between these determined. Subsequently, the matched CD40L co-culture-derived CLL transcriptomes were added to the comparison, and differentially expressed genes were identified. DESeq2 analysis was performed, and 3259 significantly differentially expressed genes (*p*.adjust ≤ 0.05 and fold-change ≥ 1.5) were identified in the migrated versus circulating cells. Surprisingly, a comparison between these and those from the matched CD40L model revealed an overlap of only 377 differentially expressed genes (Figure 1C,D), suggesting that each model induced a distinct CLL transcriptional signature with only 11.6% common differentially expressed genes. The pathway analysis revealed that these overlapping genes were mainly involved in the regulation of cell death and proliferation (Appendix A). However, there was more overlap between the significantly upregulated genes (*p*.adjust ≤ 0.05 and fold-change ≥ 1.5) between the migratory cells from our model and the lymph node-derived CLL gene expression signature identified by Herishanu et al. than the CD40L co-cultured CLL cells (46% vs. 25% respectively) (Figure 1E) [5]. This data suggests that the CD40L co-culture system is less representative of the CLL lymph node than the migratory CLL cells harvested from our circulating system.

### 3.3. Migratory CLL Cells Have a Striking Gene Set Enrichment of Adhesion, RAP1 and PI3K-AKT Signalling Pathways

The Wnt/PCP pathway has been identified as an important pathway for migration and transendothelial invasion of CLL cells [36]. We, therefore, used this as a gene set to test if the in vitro system was capturing actively migrating CLL cells. Indeed, many of the Wnt/PCP-related genes were upregulated in migratory CLL cells in comparison to the circulatory CLL cells (Appendix A), suggesting the in vitro system was successfully recreating the CLL transendothelial migratory process.

To determine the critical CLL migration gene sets, over-representation analysis (ORA) on the KEGG pathway database was performed using WebGestalt for the in vitro circulating model system RNA-seq data (Figure 1F). The migratory cells showed increased expression of genes associated with adhesion (Focal adhesion and extracellular matrix receptor interaction), a potent trigger for inside-out integrin activation and cell migration (RAP1 signalling pathway and PI3K-AKT signalling pathway). The focal adhesion gene list had an enrichment score of 2, and as FAK has previously been identified as a potential therapeutic target in several cancers [37], the focal adhesion kinase (FAK) signaling pathway was selected as a possible target to inhibit CLL migration.

### 3.4. No Clear miRNomic Role in CLL Migration Identified

MiRNomic differential expression between the circulating and migrating CLL cells was examined to understand the role of microRNAs (miRNAs) in CLL trafficking. The principal component analysis revealed a distinct circulatory and migratory CLL miRNome signature (Appendix A). Differential expression analysis identified 19 significantly upregulated miRNAs in the migratory CLL cells derived from eight individual patient samples (Appendix A). However, pathway-level analysis, combining the transcriptomic data with the miRNomic data, demonstrated that the gene targets for the 19 upregulated miRNAs were, in fact, largely upregulated (Appendix A). This suggests that the transcriptional changes observed in migratory CLL cells cannot be readily explained by reciprocal changes in miRNA expression. However, our data do not rule out a role for miRNAs in CLL migration.

### 3.5. FAK Signalling Pathway Is Upregulated during Transendothelial CLL Migration

Next, we analyzed the RNA-seq differential expression data relating to the FAK signaling pathway, presented as a Cytoscape network (Figure 2A), which indicated that downstream signaling pathways regulating cytoskeletal reorganization, cell motility, and GTPase regulation were upregulated in the migratory CLL cells, whereas the MEK-ERK pathway, which modulates cell proliferation, was not differentially expressed. The heatmap of the individual RNA-seq patient data demonstrated that the FAK signaling pathway was consistently upregulated in all the patient samples studied (Figure 2B), yielding an average *PTK2* (FAK) fold change = 4.9. This was in concordance with the increase in FAK expression observed by qRT-PCR (FAK fold change = 5.9) (Figure 2C).

### 3.6. TLR9 Stimulation Induces CLL Migration through FAK Activation and This Is Inhibited by Defactinib

Previously our group had reported that stimulating CLL cells with a TLR9 agonist, ODN2006, significantly increased CLL cell migration in 2D transwell systems [38]. We investigated if CLL migration was a FAK-dependent process by establishing whether TLR9 agonism induced p-FAK and by pharmacological inhibition of FAK using defactinib. Firstly, the p-FAK levels of primary CLL cells were determined by flow cytometry before and after TLR9 stimulation. In CLL cells from each of the three patients studied, there was a significant increase in p-FAK after 24 h of ODN2006 stimulation, which we hypothesize may be due to the iNOS/Src/FAK axis (Figure 3A) [39]. To establish whether this TLR9-induced increase in p-FAK could be reversed, we used a range of concentrations of defactinib (0.5–5 μM). In CLL cells from all three patients studied, the p-FAK levels were returned to approximate basal levels when treated with 5 µM defactinib. However, at lower concentrations (0.5 and 1 µM), no decrease in p-FAK levels was observed (Figure 3B and Appendix A). Secondly, using CLL cells from six different patients, we repeated the migration assays described above in the absence and presence of defactinib. Our results were in concordance with the CLL migratory data previously reported, with an average 1.43-fold increase in migration when stimulating with ODN2006 (Figure 3C). Furthermore, CLL cells from all six patients studied showed a marked reduction in migration in the presence of defactinib (mean = 80.7% defactinib treated vs. DMSO control; Figure 3C). However, the amount of apoptosis was only increased by 44.2% (Appendix A).

### 3.7. FAK Inhibition Results in a Heterogenic CLL Apoptotic Response

After 24 h treatment with defactinib, a heterogenous apoptotic response was observed between the CLL patient samples (Figure 3D,E). The three patient samples from the p-FAK studies were again used to assess if this heterogeneity in response was associated with p-FAK levels. We determined that there were no significant inter-patient differences in p-FAK levels either before or after TLR9 stimulation (Figure 3F,G). Additionally, there was no significant inter-patient variation in a defactinib-mediated reduction in p-FAK (Figure 3H), suggesting that the reduction in p-FAK was not a consequence of apoptosis.

### 3.8. CLL Invasion Was Inhibited by Defactinib Treatment

The data presented above show that the CLL expression of p-FAK and leukemic cell migration is inhibited by 5 µM defactinib. As well as being able to migrate, CLL cells need to be able to invade the tissue compartments to enter them. Therefore, to test the functional impact of defactinib on CLL invasion, an invasion assay plate coated with matrigel was utilized. After 24 h, in the six CLL samples, 5 µM defactinib caused a significant reduction in cell invasion (mean normalized invasion = 58% of control *p* ≤ 0.05), (Figure 4A). Importantly, no significant decrease in apoptosis was observed in the cells after the 2 h pre-treatment (Appendix A).

### 3.9. Defactinib Synergises with Ibrutinib in CLL Migration Assays

Next, we assessed the therapeutic potential for FAK inhibition alongside the BTK inhibitor ibrutinib in a preliminary study of three CLL patients. After 24 h, concentrations of defactinib below 2.5 µM had no significant effect on CLL cell migration (Figure 4B). However, 2.5 µM defactinib significantly reduced the level of CLL migration (mean migration = 56.7%). Ibrutinib was also shown to decrease CLL migration across the range of concentrations used, with a heterogeneous response between patients. Furthermore, the combination of defactinib with ibrutinib showed synergistic interactions (Bliss Score = 11.07) (Figure 4C), significantly decreasing the level of migration (mean migration = 20%) in comparison to each of the drugs alone at 2.5 µM. Importantly, the mean cell viability was not significantly different between 2.5 µM ibrutinib, defactinib, and the 2.5 µM combination, which suggests that the further decrease in CLL migration was due to FAK-dependent migration mechanisms.

## 4. Discussion

Although advancements in our understanding of CLL have facilitated the development of more effective targeted treatments, CLL remains incurable [40]. The lymphoid stromal microenvironment promotes cell survival, proliferation, and an escape from drug-induced apoptosis, so inhibition of CLL migration into these protective proliferative niches is an attractive therapeutic strategy. The effectiveness of BTK and PI3K inhibitors in tissue redistribution has revolutionized the treatment of CLL, but they are not curative, and there is heterogeneity in responses [41,42,43]. Inhibition of these kinases alone is not sufficient to completely block CLL cell trafficking, and so in this study, we set out to further elucidate the migratory process and identify potential novel targets.

To do this, we utilized a dynamic circulatory model, previously described by Walsby et al. [27], which circulated CLL cells through endothelium-lined hollow fibers under physiologically relevant shear forces simulating the transient interaction of the CLL cells with endothelial cells. Performing RNA-seq on the migratory population of CLL cells isolated from the system and comparing it to the circulating population, allowed us to investigate the transcriptional drivers of transendothelial migration. Consistent with previous studies, an upregulation of the Wnt/PCP pathway from the CLL migratory population was observed, further validating the use of a circulatory model for recapitulating CLL transendothelial migration [36].

We have previously shown that migratory CLL cells have a phenotype strikingly similar to those that reside in the lymph nodes [28]. A direct comparison between a commonly used CD40L fibroblast co-culture model and our in vitro migration system was performed to assess which model could more accurately recreate the CLL lymph node transcriptomic signature. Firstly, CLL cells co-cultured on CD40L fibroblast exhibited an activated phenotype with an increase in CD69 expression. However, in contrast to previously published data on lymph node fine-needle aspirates, CD38 and CXCR4 were not significantly altered [28]. This is likely due to the reductionist nature of the co-culture model, e.g., the absence of CXCL12 stimulation. In contrast, migratory cells from our circulatory model, which incorporates shear force, endothelial cell interaction, and CXCL12 chemotaxis, were shown to more accurately represent the transcriptional signatures of LN resident CLL cells [5]. This suggests that part of the lymph node gene signature is induced during the process of migration rather than by the microenvironment of the lymphoid niche.

To assess the upregulated gene sets during CLL migration, overrepresentation analysis of the migrating CLL cells was performed, which identified focal adhesion. Furthermore, the FAK signaling pathway was shown to be highly upregulated, so FAK was selected as a possible target for the inhibition of CLL migration. FAK, a protein tyrosine kinase that plays a key role in integrin signaling, mediating cell adhesion, cytoskeletal reorganization, as well as cell proliferation and survival, has been reported to be overexpressed and activated in several other cancers [37,44,45]. Moreover, it has previously been shown to participate in CXCL12-induced activation of the PI3K-AKT pathway [46], which in this study, was overrepresented in migrating CLL cells. Additionally, our findings show that the RAP1 signaling pathway, a potent trigger for inside-out integrin activation, was upregulated. Furthermore, a previous study reported that CXCL12-mediated RAP1 activation was absent in FAK-deficient acute lymphoblastic leukemia cells [47], indicating that FAK is an important regulator of the CXCL12-induced integrin activation pathway, and making it an attractive target for the inhibition of CLL migration.

In line with this hypothesis, we demonstrated that CLL cells treated with defactinib have significantly reduced migratory and invasive potential, indicating that FAK inhibition could effectively reduce CLL lymph node ingress. Importantly, the reduction in migration could not be explained by defactinib-induced apoptosis, indicating that the mechanism of FAK inhibition was due to the on-target effects of the drug. This observed defactinib-induced apoptosis has been reported in several other cancers where it has been linked to a reduction in NF-κB and PI3K-AKT signaling [48,49]. As our CD40L fibroblast co-culture model highlighted, NF-κB signaling is induced by activated T cells in the lymph node microenvironment, and therefore, FAK inhibition could also play an important role in inhibiting NF-κB and PI3K-AKT induced pro-survival signaling. In support of this, the CD40L fibroblast co-culture model was shown to induce an increase in the anti-apoptotic BCL-2 members, MCL-1, BCL-XL, and BCL2A1, all of which are NF-κB target genes associated with venetoclax resistance [50]. As FAK inhibition has been demonstrated to reduce MCL-1 and BCL-XL expression, and synergistically induce apoptosis in AML when used alongside venetoclax, this combination could be an effective therapeutic strategy by both promoting apoptosis and inhibiting migration in CLL [48].

Ibrutinib has been shown to produce a transient lymphocytosis which is due to the efflux of CLL cells from the lymphoid organs into the peripheral blood [41]. However, with the emergence of ibrutinib resistance, and CLL cases expressing high levels of the adhesion molecule, CD49d, typically failing to display ibrutinib-induced lymphocytosis, there is an urgent need to develop new combination therapies for these subsets of patients [23]. Another potentially effective combinatorial treatment for CLL could be FAK inhibition alongside the BTK inhibitor ibrutinib. Rudelius et al. demonstrated that ibrutinib and FAK inhibition were highly synergistic in mantle cell lymphoma, reporting complete abrogation of NF-κB signaling pathway, even in ibrutinib resistant cells [47]. Our preliminary data were in concordance with this study, with our results indicating that there is a synergistic interaction between defactinib and ibrutinib in CLL migration inhibition. This highlights the potential for this combinatorial strategy to maximize the inhibition of CLL migration and induce apoptosis in this largely incurable disease.

## 5. Conclusions

In conclusion, migratory CLL cells from our circulating model system more accurately represented the transcriptional signatures of LN resident CLL cells than co-culture with the commonly used 2D CD40L fibroblast model. This suggests that the process of migration constitutes part of the lymph node gene signature. Furthermore, the number of upregulated genes observed in the CLL cells undergoing transendothelial migration, highlights the complexity of the process. From the overrepresented gene sets identified, we selected the FAK signaling pathway due to its key regulatory role in cell adhesion and migration and promise as a targetable therapeutic agent in several cancers. Inhibition of FAK in primary CLL samples was found to effectively reduce both CXCL12 induced migration and invasion in vitro. This highlights FAK inhibition as an exciting potential combinational therapy for CLL.

## Figures and Tables

**Figure 1 cancers-14-01600-f001:**
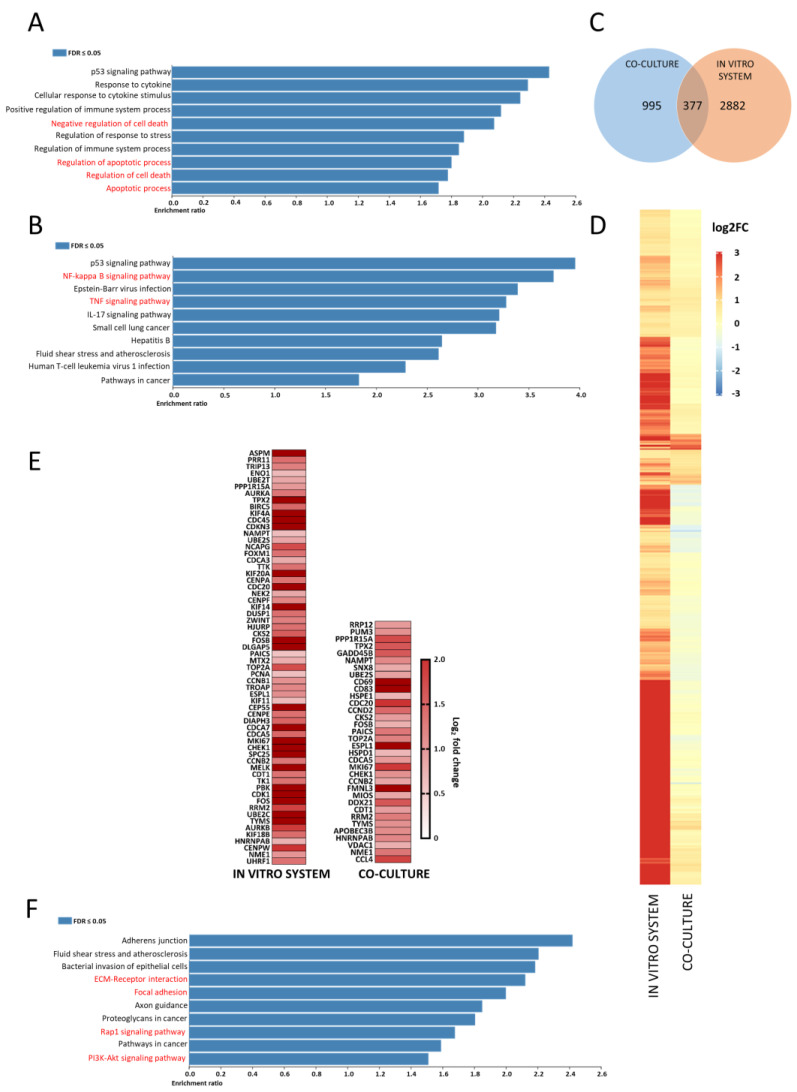
Fibroblast co-culture and endothelial cell in vitro system produce distinct transcriptomic signatures. (**A**) The top 10 overrepresented pathways in the differentially expressed gene list for CD40L-fibroblast co-cultured cells in the biological processes and (**B**) the KEGG pathway databases. (**C**) Shows the overlap in upregulated genes between CD40L co-culture and in vitro circulatory system. (**D**) Shows a heatmap displaying the significantly upregulated genes in the in vitro circulatory system and the corresponding expression in the co-culture system. (**E**) Shows a heatmap representing the overlap between the differentially upregulated genes between migratory cells in the circulating system, the CD40L co-culture system and the 134 previously published genes upregulated in lymph node resident CLL cells. (**F**) The top 10 overrepresented pathways in the differentially expressed gene list for in vitro system migratory cells in the KEGG pathway database. Red text represents gene sets of interest.

**Figure 2 cancers-14-01600-f002:**
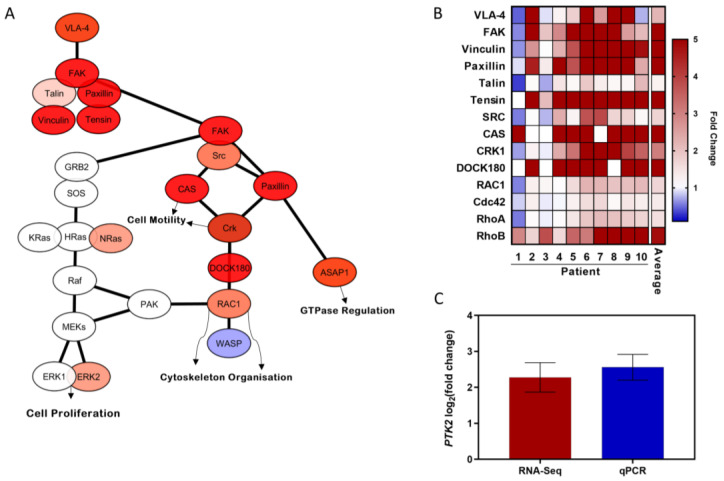
Migratory CLL cells harvested from our in vitro circulating system upregulate the FAK signaling pathway in comparison to CLL circulatory cells. (**A**) RNA-seq differential expression overlayed on a FAK signaling pathway Cytoscape network. (**B**) Heatmap of 14 differentially upregulated FAK signaling pathway genes from migratory CLL cells. (**C**) For validation, FAK gene expression from the RNA-seq data (*n* = 10) was compared to the qPCR data (*n* = 4) as assessed by a TaqMan assay.

**Figure 3 cancers-14-01600-f003:**
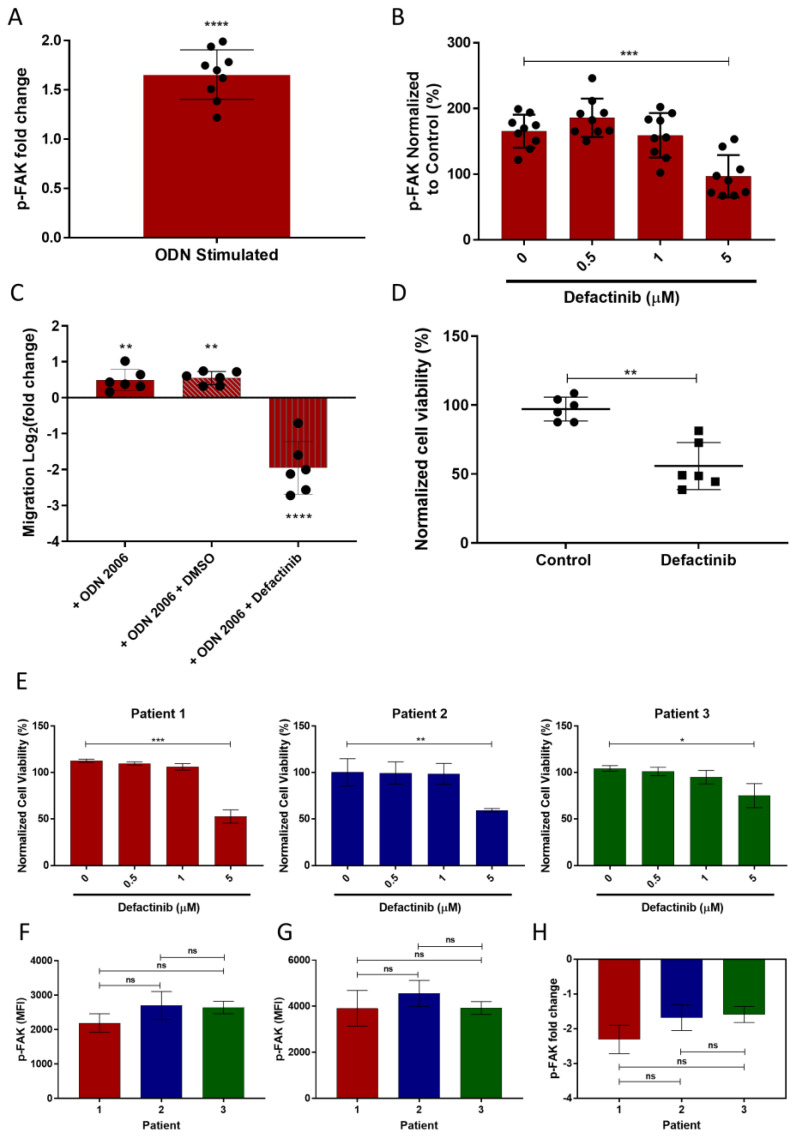
Stimulating CLL cells through TLR9 causes an increase in p-FAK and migration, which was abrogated by FAK inhibition. (**A**) PBMCs from three different CLL patients were incubated with or without ODN2006 for 24 h in triplicate and the p-FAK levels were assessed by flow cytometry. The mean fluorescence intensity was determined for both groups and the fold change in p-FAK was calculated. (**B**) Alongside TLR9 stimulation, CLL cells were treated with a range of defactinib concentrations for 24 h and the subsequent p-FAK levels were measured. (**C**) CLL cells from six patients were incubated with or without ODN2006 and treated with a vehicular control or defactinib (5 µM) overnight before transferring into transwell migration chambers and allowed to migrate towards a CXCL12 gradient for 4 h. The migrated cells were quantified by volumetric counting. (**D**) Defactinib induced apoptosis after 24 h treatment in the six TLR9 stimulated CLL patient samples, as assessed by 7AAD/Annexin V staining. (**E**). PBMCs from three different patients with CLL were incubated overnight with or without stimulation with ODN2006 with or without co-treatment with defactinib (0–5 μM). The percentage cell viability for the CLL cells with and without defactinib treatment for 24 h was determined by Annexin V/7-AAD staining (**F**) The levels of p-FAK were assessed by flow cytometry before ODN stimulation (**G**) Post-ODN stimulation. (**H**) p-FAK fold change after 24 h defactinib treatment in ODN stimulated CLL cells. **** *p* ≤ 0.0001 *** *p* ≤ 0.001, ** *p* ≤ 0.01, * *p* ≤ 0.05.

**Figure 4 cancers-14-01600-f004:**
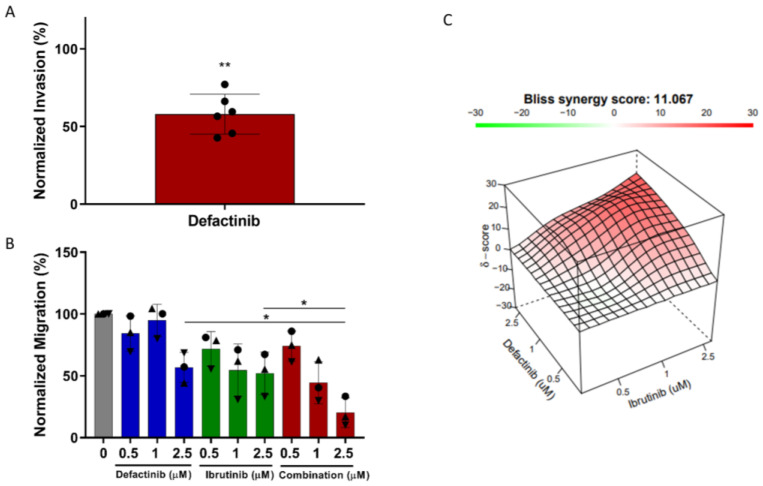
Pharmacological inhibition of FAK activation reduces CLL invasion and is synergistic with ibrutinib. (**A**) PBMCs from six patients were pre-treated with defactinib for 2 h and then transferred into a BioCoat matrigel invasion chamber and allowed to invade towards a CXCL12 gradient for 24 h. The migrated cells were quantified by volumetric counting. (**B**) PBMCs from three patients were incubated with defactinib (0.5, 1, 2.5 µM), ibrutinib (0.5, 1, 2.5 µM), or a combination of both (molar ratio 1:1) overnight before transferring into transwell migration chambers and allowed to migrate towards a CXCL12 gradient for 4 h. The migrated cells were quantified by volumetric counting. Shapes represent different patients (**C**) The synergy between defactinib and ibrutinib was determined using the SynergyFinder software (https://synergyfinder.fimm.fi, accessed on 21 November 2021). * *p* ≤ 0.05,** *p* ≤ 0.01.

## Data Availability

The data generated in this publication has been deposited in NCBI’s Gene Expression Omnibus and will be accessible through GEO Series accession number GSE198456 (https://www.ncbi.nlm.nih.gov/geo/query/acc.cgi?acc=GSE198456, accessed on 30 January 2022).

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
