# Peer review of "Elucidation of Focal Adhesion Kinase as a Modulator of Migration and Invasion and as a Potential Therapeutic Target in Chronic Lymphocytic Leukemia"

_cancers, 2022, doi:10.3390/cancers14071600_

Round 1
Reviewer 1 Report
In this manuscript entitled “Elucidation of focal adhesion kinase as a modulator of migration and invasion and as a potential therapeutic target in chronic lymphocytic leukemia”, Burley and colleagues investigated the molecular mechanisms involved in CLL cells migration. This topic is critical for CLL since a “disease reservoir” is hosted within the lymph nodes where neoplastic cells are protected from the effects of therapies. Highlighting the mechanisms chemotaxis towards lymph nodes rely on are a necessary step to improve current treatment options.
The technical approaches adopted by the Authors are based on the use of a hollow fiber circulatory system to better mimic in vivo CLL cells migration and on a gene expression profiling to identify differentially expressed genes.
The topic is of high interest and the work has several merits. However, there are also some issues that need to be address before the manuscript can be considered for publication.
Major points
- In paragraph 3.6 of the Results section, the Authors showed that Defactinib is able to reduce migration of CLL in response to ODN2006 at the 5 microM dose, while being ineffective at lower doses. Moreover, they demonstrated that this inhibitor is inducing apoptosis after 24-hour treatment. To better interpreter these results and rule-out that the reduced migration is due to apoptosis of CLL cells, a cell viability assay at 4-hour treatment must be shown. Moreover, to better sustain these results, migration and apoptotic assays must be performed in a larger cohort of patients (the actual cohort made of 3-4 patients present some degree of variability and it may be too small to reach statistical significance and obtain consistent results).
- In paragraph 3.8 of the Results section, the Authors evaluated the impact of Defactinib on CLL cells invasion capacity. The data presented are quite confusing: it is not clear whether results refer to only a single CLL patient, as stated in Figure S8 that show that the 24-hour viability of a specific CLL case or whether the Authors considered a bigger cohort of patients. If the latter is the case, number of patients considered must be indicated. Moreover, these data appeared to be in contrast with results presented in the previous section showing that Defactinib was able to induce apoptosis in CLL cells after 24-hour treatment. An additional point to be clarified is whether CLL cells were treated with Defactinib throughout the duration of the invasion assay (24 h) or if the inhibitor was used only for a 2-hour pre-treatment then wash out prior to start the invasion assay.
- Since the Authors identified FAK and the PI3K/anti-apoptotic pathways as upregulated in migrating CLL cells, it would be interesting to study whether a combination inhibitory approach targeting FAK and PI3K (e.g., using Duvelisib) or FAK and Bcl-2 (e.g., using Venetoclax) may result in a more pronounced decrease in migration, invasion, and cell viability.
Minor points
- In paragraph 3.2 of the Results section, Authors said that the overlap between the migratory cells profile (from their model) and the lymph-node derived CLL cells signature overlap for the 46%, which is however higher that the results obtained using the CD40L-based co-colture system. How do the Authors explain the difference in terms of gene expression profile obtained by their system and previously published data from lymph-node derived CLL cells? Is the difference due to the fact that they are looking to circulating CLL cells while published data consider also neoplastic cells that were strictly in contact to other cellular populations present in the lymph node for a longer period of time? Please provide a possible explanation.
- Figure legends need some minor revisions to avoid typos and misleading/confusing sentences. Moreover, letters referring to the different panels need to be placed before the legend description (e.g., Figure 3 legend “(D) Defactinib induced apoptosis after 24h treatment 4 TLR9 stimulated CLL patient samples, as assessed by 7AAD/ Annexin V staining. PBMCs from 3 different patients with CLL were incubated with or without ODN2006 overnight and treated alongside with a range of defactinib concentrations (E) The % cell viability for the CLL cells with and without defactinib treatment for 24h was determined by 7AAD/Annexin V staining. The levels of p-FAK were assessed by flow cytometry (F) Before ODN stimulation (G) Post-ODN stimulation.”
- Figure 1 quality must be improved. Moreover, Panels A, showing the biological processes, and B, showing the KEGG pathways, are reporting the same data and appeared to be redundant. Authors should consider removing one of the two.
- Figure S6 refers to hierarchical analysis of differentially expressed miRNA between 8 paired CLL samples, while in the text (lane 281) Authors indicate that the analysis was performed on 10 patients. Please clarify.
- Since Authors are looking at the expression profile of CLL cells following migration in different conditions, a table showing the cohort of patients considered and their phenotypic characteristics, including CXCR4, TLR9, CD49d, CD38 expression (as % of positive cells and MFI), must be added in the manuscript.
Reviewer 2 Report
In this manuscript, the authors applied two distinct in vitro CLL models to study the molecular processes of CLL cells migration and invasion. By analysis of CLL transcriptomics and phenotypic changes, they found that pro-survival, anti-apoptotic, NF-κB and TNF signaling pathways are involved in CLL cells migration and invasion. They finally identified a key modulator named focal adhesion kinase (FAK). Inhibition of FAK using a pharmacological inhibitor - defactinib significantly reduce CLL cells migration and invasion, indicating FAK to be a potential therapeutic target in CLL treatment.
The manuscript overall are well written, the experiments are all well-designed and results are well analyzed and I would suggest the acceptance of the manuscript after some minor revisions.
- Materials and Methods 2.2 In vitro circulatory system. A hollow fiber bioreactor system was used to mimic the circulation of CLL cells. As this system is new to me and should also be new to most of reviewers, it is suggested the authors draw a schematic figure to illustrated how the system works.
- In the manuscript, the gene enrichment assays are based on differential expressions between circulating and migrating CLL cells. Could the authors explain why such kind of comparisons are applied in the studies.
- Figure 2B, in heatmap RAC gene expression was highlighted, is it RAC1 or RAC2?
- Please provide the original Flow graph of Annexin-V and 7-AAD assay, and also Flow graph of intracellular stain of p-FAK.
